# Effect of *Pennisetum giganteum* z.x.lin mixed nitrogen-fixing bacterial fertilizer on the growth, quality, soil fertility and bacterial community of pakchoi (*Brassica chinensis* L.)

**Yulei Jia**[1,2☯], **Zhen Liao**[1,2☯], **Huifang Chew**[1,2], **Lifang Wang**[1,2], **Biaosheng Lin**[1,2,3], **Chaoqian Chen**[1,2], **Guodong Lu**[1,2], **Zhanxi Lin**[1,2]*

1 College of Life Science, Fujian Agriculture and Forestry University, Fuzhou, PR China, 2 National Engineering Research Center of Juncao Technology, Fuzhou, PR China, 3 College of Life Science, Longyan University, Longyan, PR China

☯ These authors contributed equally to this work.
* lzxjuncao@163.com

**Data Availability Statement:** All relevant data are within the manuscript and its Supporting Information files.

## Abstract

Biofertilizer plays a significant role in crop cultivation that had reduced its inorganic fertilizer use. The effects of inorganic fertilizer reduction combined with *Pennisetum giganteum* z.x. lin mixed nitrogen-fixing biofertilizer on the growth, quality, soil nutrients and diversity of the soil bacterial community in the rhizosphere soil of pakchoi were studied. The experiment composed of 6 treatments, including CK (no fertilization), DL (10% inorganic fertilizer reduction combined with *Pennisetum giganteum* z.x.lin mixed nitrogen-fixing biofertilizer), ZL (25% inorganic fertilizer reduction combined with *Pennisetum giganteum* z.x.lin mixed nitrogen-fixing biofertilizer), SL (50% inorganic fertilizer reduction combined with *Pennisetum giganteum* z.x.lin mixed nitrogen-fixing biofertilizer), FHF (100% inorganic fertilizer) and JZ (100% inorganic fertilizer combined with sterilized *Pennisetum giganteum* z.x.lin mixed nitrogen-fixing biofertilizer). Compared with conventional fertilization, the 25% reduction in chemical fertilizer applied with the *Pennisetum giganteum* mixed nitrogen-fixing biofertilizer resulted in higher plant height, plant weight, chlorophyll content, soluble protein content, soluble sugar content, vitamin C content, alkali hydrolyzed nitrogen content, available phosphorus content, available potassium content and organic matter content in pakchoi, and these variables increased by 11.81%, 8.54%, 7.37%, 16.88%, 17.05%, 23.70%, 24.24%, 36.56%, 21.09% and 19.72%, respectively. In addition, the 25% reduction in chemical fertilizer applied with the *Pennisetum giganteum* mixed nitrogen-fixing biofertilizer also had the lowest nitrate content, which was 53.86% lower than that with conventional fertilization. Different fertilizer treatments had a significant effect on the soil bacterial community structure. Compared with conventional fertilization, the coapplication of *Pennisetum giganteum* z.x.lin mixed nitrogen-fixing biofertilizer and inorganic fertilizer significantly increased the relative abundance of Proteobacteria and Actinobacteria in the soil. The results of the redundancy analysis (RDA) showed that soil organic matter, alkali-hydrolyzed nitrogen, available phosphorus, available potassium, pH and water content had a specific impact on the soil bacterial community. Among the factors, soil water content was the main factor affecting the soil

**Funding:** This work was supported by the Central Leading Local Science and Technology Development Special Fund Project of China under grant number 2018L3003 and the Ministry of Agriculture, Grass and Ecology Industry and the Ministry of Construction and Collaborative Innovation Center, China under grant number [2018]126.

**Competing interests:** The authors have declared that no competing interests exist.

bacterial community, followed by soil organic matter, soil pH, available potassium, soil available phosphorus and soil alkali-hydrolyzed nitrogen.

## Introduction

Soil microorganisms are an important part of the soil ecosystem [1], participating in the decomposition of organic matter, nutrient element cycles and energy conversion [2–4]. These microorganisms play an important role in maintaining the productivity, function and stability of the ecosystem [5–6], which is a key indicator for measuring soil quality and productivity [7]. Bacteria are the most abundant and largest group of microorganisms, usually accounting for 70%~90% of soil microorganisms. Bacteria have the richest genetic diversity and can effectively promote the decomposition of organic matter and release of nutrients, participating in nutrient cycling processes such as carbon and nitrogen cycling and maintaining the energy flow and material cycles of the ecosystem [8–10].

With the development of modern agriculture in China, inorganic fertilizer plays a very important role in agricultural production. However, the long-term unsustainable application of inorganic fertilizers has resulted in serious problems to the ecological environment that occur daily. A large number of studies have shown that the amount of fertilizer applied in China has exceeded the optimal amount economically and has resulted in losses of economic benefits to farmers [11]. At the same time, excessive application of inorganic fertilizer has also caused critical environmental pollution. Scientific research has confirmed that the overuse of inorganic fertilizers has become the main source of agricultural pollution [12–14]. Biofertilizers are products containing living cells of different types of microorganisms that have the ability to convert nutritionally important elements from unavailable to available forms through biological processes [15]. Biofertilizer can not only promote plant growth and development and improve the stress resistance of crops and the quality of agricultural products [16–17] but also improve soil fertility, fertilizer utilization efficiency and soil microbial community structure [18–21].

Pakchoi (*Brassica chinensis L.*) is cultivated within a large northern to southern range in China [22]. *Pennisetum giganteum* z.x.lin belongs to the phylum angiospermae, class monocotyledons, family gramineae and genus *Pennisetum*. It is a typical C4 plant that is suitable for growing in tropical, subtropical and temperate zones [23]. *Pennisetum giganteum* z.x.lin has high nutritional value, good palatability and wide application. It can not only replace sawdust to cultivate edible and medicinal mushrooms, addressing the conflict between forest and mushroom industry development but also be used as animal forage, and it has wide application in ecological management [24]. An endophytic diazotroph is a kind of microorganism that colonizes healthy plants and combines with host plants for nitrogen fixation. It plays an important role in nitrogen fixation, biological control and plant growth [25]. Peng [26] et al. found that the light absorption and growth rate of rice leaves increased by 12% and the yield increased by 16% after inoculation with endophytic diazotrophs. Govindarajan [27] et al. inoculated Burkholderia MG43 into sugarcane and found that this approach could replace half of nitrogen fertilizer and save 70 kg ha-1 nitrogen fertilizer. At present, there are many studies on the endophytic nitrogen-fixing bacteria of Gramineae and crops, but there are few studies on the endophytic nitrogen-fixing bacteria of *Pennisetum giganteum* and its nitrogen-fixing bacteria fertilizer. To provide a theoretical basis for the scientific and appropriate application of *Pennisetum giganteum* z.x.lin mixed nitrogen-fixing biofertilizer and partial replacement of

chemical fertilizer, this experiment was carried out to study its effect on the growth and quality of pakchoi, the physical and chemical properties of soil and the diversity of the bacterial community.

## Materials and methods

### Experimental site description

The experiment was conducted in August 2018 at the greenhouse of the National Engineering Research Center of JUNCAO Technology base, Fujian Agriculture and Forestry University, Fuzhou, China. The soil organic matter content was 31.14 g/kg, the alkali-hydrolyzed nitrogen content was 36.0 mg/kg, the available phosphorus content was 63.1 mg/kg, the available potassium content was 115.5 mg/kg, and the pH value was 5.26.

### Preparation of test materials

The *Kosakonia radicincitans* nitrogen-fixation strain was isolated from *Pennisetum giganteum* z.x.lin and preserved in the laboratory. The sterilized samples of macromycorrhizal roots were placed into a sterilized mortar and cut with sterile scissors, and then, the proper amount of PBS buffer to grind the roots was added. The extract (0.1 mL) was coated on an Ashby solid medium plate, and the culture was inverted at 28~30°C for 2–3 days to observe colony growth. Then, the single bacteria growing on the Ashby solid medium plate was inoculated to the Nfb solid medium again for rescreening. The strains that could grow on the Nfb medium were selected, the color of the medium changed from blue to green, and the strains were inoculated again on the Ashby solid medium. This process was repeated three times. Finally, the selected colonies were inoculated on the Nfb slant medium and saved for future use and identification [28]. *Bacillus mucilaginosus* was isolated from the rhizosphere soil of *Pennisetum giganteum* and preserved in the laboratory. After gradient dilution, the soil samples were coated on silicate bacteria culture medium and cultured in a constant temperature incubator at 30°C for 48 h. According to the growth speed and morphology of the colony, a clear colony with mucilaginous protuberance was removed from the separation plate and then separated on the solid plate until a pure culture was obtained [29]. The pakchoi variety was No.2 *Brassica chinensis* from New Zealand that was bought from the local market. Inorganic fertilizer (N:P:K = 18:6:6) was bought from Fujian AoLiGaoTa Fertilizer Co., Ltd. The stems and leaves of *Pennisetum giganteum* z.x.lin were selected from the jointing stage or mature stage, crushed, dried at a low temperature and passed through a 100 mesh sieve. The waste mushroom substrates of *Ganoderma lucidum* were selected and cultivated with Juncao grasses, dried and crushed at a low temperature and passed through a 100 mesh sieve. The nutrient solution formula consisted of brown sugar 30–50 g/L, $MgSO_4$ 0.5–1.5 g/L, and calcium superphosphate 10–20 g/L.

### Preparation of *Pennisetum giganteum* z.x.lin mixed nitrogen-fixing biofertilizer

The activated *Kosakonia radicincitans* and *Bacillus mucilaginosus* were inoculated into an Luria-Bertani (LB) liquid medium at a ratio of 1:1, the culture oscillated at 150–180 r/min at 30°C until logarithmic growth. In a polyethylene film fermentation bag with a breathing valve, 150–200 mL bacterial solution, 650–750 g of dry *Pennisetum giganteum* z.x.lin, 250–350 g waste mushroom substrates of *Ganoderma lucidum*, and 100–150 mL nutrient solution were mixed, and fermentation occurred at 25–32°C for 5–10 days. Fermentation ended if the material became soft and dark. Compound microbial fertilizer was tested according to NY411-2000 nitrogen-fixing bacteria fertilizer [28].

## Experimental test design

The experiment of inorganic fertilizer reduction combined with *Pennisetum giganteum* z.x.lin mixed nitrogen-fixing biofertilizer was carried out. Six treatments were set up, and three plots with a replication of each treatment were arranged in completely random groups. The area of each plot was 12 m², with a total of 18 plots. The amount of inorganic fertilizer applied was 525 kg ha⁻¹ according to the recommended amount of fertilizer applied locally. The proportion of fertilizer applied is shown in Table 1.

## Sampling and analysis

On the harvest day, the whole plant sample was collected. The root soil was washed, plant height was measured with a ruler, and the fresh weight was measured by electronic balance. Chlorophyll content was determined by a SPAD-502 Plus chlorophyll content analyzer. The soluble protein content was determined by Coomassie brilliant blue colorimetry [30]. The soluble sugar content was determined by anthrone colorimetry [31]. The vitamin C content was determined by the 2,6-dichlorophenol-indophenol method [32]. The nitrate content was determined by nitrate-nitrogen colorimetry [33].

The soil samples were collected at depths of 0–20 cm. The "S" method was used in each plot to collect the rhizosphere soil samples of the pakchoi at 5 random points, and the samples were fully mixed as a soil sample. The method was repeated in each plot three times. Fresh soil samples were divided into two parts. One part was brought back to the laboratory for cryopreservation at -80 °C for soil microbial sequencing analysis. The second part was brought back to the laboratory for indoor air-drying to determine the soil physical and chemical properties. The soil organic matter was determined by the potassium dichromate volumetric dilution method. In a 500 mL triangular flask, 0.5000g of the soil sample was accurately weighed; then, 1 mol/L (1/6 $K_2Cr_2O_7$) solution was added to the soil sample to ensure it mixed evenly; and finally, 20ml of $H_2SO_4$ was added, and the triangular flask was slowly rotated to ensure the reagent and soil fully mixed to oxidize the soil organic matter. The alkali-hydrolyzed nitrogen was determined by the alkali-hydrolyzed diffusion method. This method involved weighing 2.00 g of air-dried soil sample, passing it through a no. 18 sieve, putting the material into a diffusion dish, adding 1 mol/L NaOH solution to 10.0 ml of hydrolyzed soil, and converting the hydrolyzed nitrogen alkali hydrolyzed into $NH_3$. The available phosphorus was determined by the molybdenum blue colorimetric method. This method involved weigh 2.5 g of the air-dried soil sample, passing the sample through a 20 mesh sieve into a 150 mL triangular flask and adding 50 mL of 0.5 mol/L $NaHCO_3$ solution to ensure the sample was fully mixed with the

**Table 1. The proportion of fertilizer in the different fertilizer treatments.**

| Sample ID | Treatment groups | *Pennisetum giganteum* z.x.lin mixed nitrogen-fixing biofertilizer application (kg ha⁻¹) | Inorganic fertilizer application (kg ha⁻¹) |
|---|---|---|---|
| CK | No fertilization | 0 | 0 |
| DL | 10% inorganic fertilizer reduction combined with *Pennisetum giganteum* z.x.lin mixed nitrogen-fixing biofertilizer | 750 | 472.5 |
| ZL | 25% inorganic fertilizer reduction combined with *Pennisetum giganteum* z.x.lin mixed nitrogen-fixing biofertilizer | 750 | 393.75 |
| SL | 50% inorganic fertilizer reduction combined with *Pennisetum giganteum* z.x.lin mixed nitrogen-fixing biofertilizer | 750 | 262.5 |
| FHF | Conventional fertilization | 0 | 525 |
| JZ | Conventional fertilization combined with sterilized *Pennisetum giganteum* z.x.lin mixed nitrogen-fixing biofertilizer | 750 | 525 |

soil to extract the available phosphorus in the soil. The available potassium was determined by the flame photometric method. This method involved weighing 5.00 g of the air-dried soil sample, passing it through a 1 mm sieve into a 100 mL triangular flask and adding 50 mL of a 1 mol/L neutral $NH_4OAc$ solution to ensure the sample was fully mixed with the soil for extraction of available potassium in the soil. The oil pH was determined by the potentiometric method [34].

## Extraction of soil genomic DNA, amplification of the 16S rDNA v3-v4 region and high-throughput sequencing

Total genomic DNA from the soil was extracted using a Mobio PowerSoil® DNA Isolation Kit (Mobio, USA). After the extraction of genomic DNA, 1% agarose gel electrophoresis was used to detect the extracted genomic DNA. Using extracted soil genomic DNA as the template, the 16S rDNA v3-v4 region was selected as the amplified fragment by PCR amplification, followed by high-throughput sequencing. Primer sequences were 336F (5'—gtactcctacgggaggcagca-3') and 806R (5'—gtggactachvgggtwtctaat-3') [35]. The PCR system (25 μL) involved the following: 30 ng DNA samples, forward primer (5 μmol/L) 1 μL, reverse primer (5 μmol/L) 1 μL, BSA (2 ng/μL) 3 μL, 2 x Taq Plus Master Mix 12.5 μL, and dd $H_2O$ 7.5 μL. The PCR conditions were as follows: predenaturation at 94°C for 5 min, denaturation at 94°C for 30 s, annealing at 50°C for 30 s, extension at 72°C for 60 s, and 30 cycles. Finally, the process was extended at 72°C for 7 min. The amplification results were subjected to 2% agarose gel electrophoresis, and the PCR products were recovered using an AxyPrepDNA gel recovery kit (AXYGEN). The PCR products were eluted by Tris_HCl, detected by 2% agarose electrophoresis and then sequenced by the Illumina Miseq sequencing platform from the Beijing Ollwegene Technology Co., LTD.

## Processing of sequencing data

First, the double-ended sequence data obtained by the Illumina Miseq sequencing slice pairs of reads were joined together into a sequence according to the overlap relationship between the PE reads. At the same time, the quality of the reads and the effect of splicing were filtered by quality control, and the samples were effectively distinguished according to the sequence at the ends of the fore and aft barcode and primer sequence to calibrate sequence direction, namely, to optimize the data. Then, OTU clustering analysis was performed on the valid data in the samples at the 97% level. Based on the results of the OTU clustering analysis, a multiple diversity index analysis and sequencing depth detection were performed on the OTUs. Based on taxonomic information, statistical analysis of bacterial community structure was carried out at various classification levels.

The data were statistically analyzed by graphpad prism 5.01 software, Data were statistically evaluated using analysis of variance (ANOVA) tests. All statistical analyses were considered significant at the $P < 0.05$ level. The results are presented as the mean ± standard deviation (SD). Different small letters are significantly different at the 0.05 level under different fertilization treatments.

## Results

### Growth and chlorophyll content of pakchoi under different fertilizer treatments

Compared with other treatments, in treatment ZL, the height of pakchoi significantly increased and was 11.81% higher than that in treatment FHF (Fig 1a). In terms of fresh weight

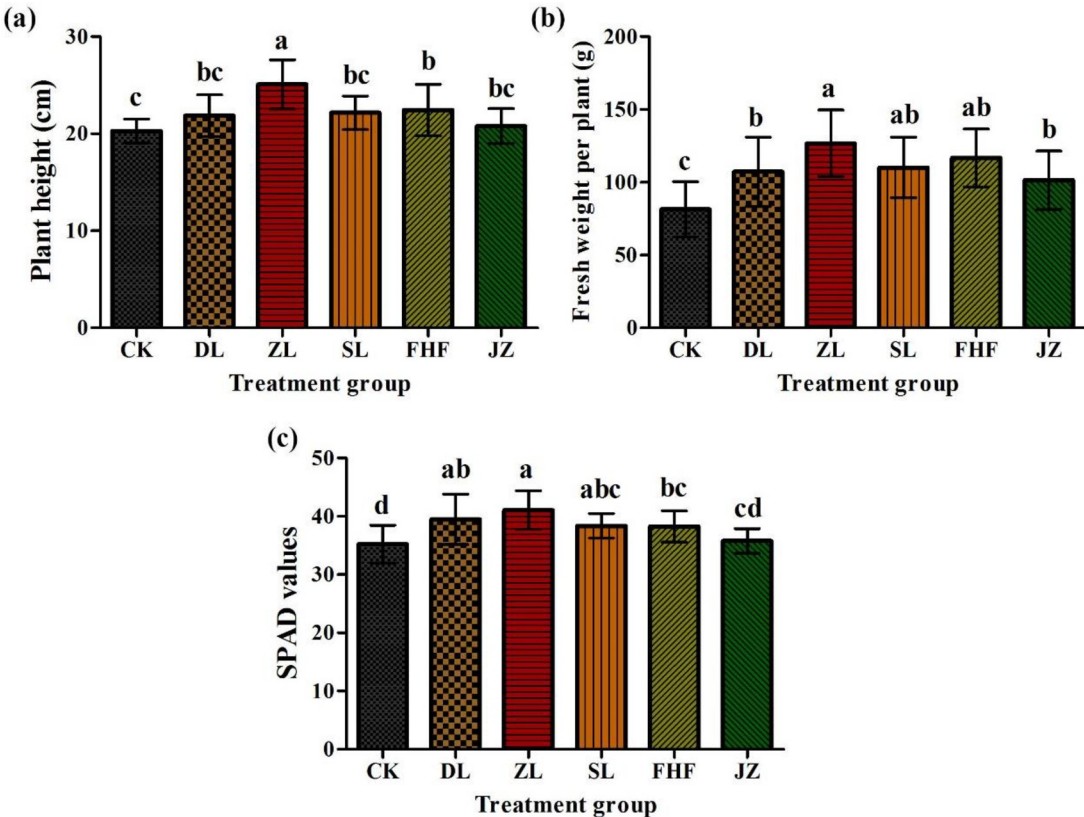

**Fig 1. Effect of different fertilizer treatments on pakchoi: (a) plant height, (b) fresh weight per plant, and (c) chlorophyll content.**

per plant, there was no significant difference between treatment ZL and treatments SL and FHF. However, there was a significant difference among treatments DL, JZ and CK (Fig 1b). Among them, treatment ZL had the highest per plant fresh weight, which was 8.54% higher than that of treatment FHF (Fig 1b). In the chlorophyll fraction, there was no significant difference in chlorophyll content between treatment ZL and treatments DL and SL. However, compared with treatments FHF, JZ and CK, there were significant differences (Fig 1c). The highest chlorophyll content was found in treatment ZL, which was 7.37% higher than that in treatment FHF.

## Nutrient concentration

Compared with treatment FHF, treatments DL, ZL, and SL had higher soluble protein, soluble sugar, and vitamin C contents in pakchoi. Among them, treatment ZL had the highest soluble protein, soluble sugar, and vitamin C contents, and they increased by 16.88%, 17.05% and 23.70%, respectively, compared with those of treatment FHF. However, the nitrate content under treatment ZL was the lowest, which was 53.86% lower than that of treatment FHF (Table 2).

## Physicochemical properties of pakchoi rhizosphere soil

The physical and chemical properties of the soil are shown in Table 3. The results show that the content of organic matter, alkali-hydrolyzed nitrogen, available phosphorus and available

**Table 2. Effect of different treatments on pakchoi nutrients.**

| Treatments | Soluble protein (mg/g) | Soluble sugar (mg/g) | Vitamin C (µg/g) | Nitrate (µg/g) |
|---|---|---|---|---|
| CK | 25.57±0.77[d] | 16.60±1.43[c] | 180.98±14.03[c] | 175.06±16.35[ab] |
| DL | 29.84±0.94[abc] | 20.45±0.75[ab] | 232.96±22.47[ab] | 147.71±18.89[b] |
| ZL | 32.41±0.83[a] | 22.11±1.34[a] | 244.66±14.29[a] | 143.12±16.37[b] |
| SL | 30.90±1.16[ab] | 21.02±0.78[ab] | 225.21±16.80[abc] | 163.46±14.49[b] |
| FHF | 27.73±1.32[cd] | 18.89±0.75[bc] | 197.79±17.61[bc] | 220.20±20.34[a] |
| JZ | 29.11±0.73[bc] | 19.38±1.47[abc] | 204.03±16.83[abc] | 214.69±17.62[a] |

The data are presented as the mean ± standard deviation (SD). Different letters in the same column mean significant difference at 0.05 level.

potassium in the soil increased due to conventional fertilization and application of *Pennisetum giganteum* z.x.lin mixed nitrogen-fixing biofertilizer in the six treatments. The content of alkali-hydrolyzed nitrogen in treatment ZL was the highest, which was not significantly different from that in treatments DL, SL, FHF and JZ but significantly different from that in treatment CK. The soil available phosphorus content in treatment ZL was the highest, which was not significantly different from that in treatment SL and was significantly different from that in treatments DL, FHF, JZ and CK. The available potassium content in the soil in treatment ZL was the highest; there was no significant difference between treatments DL, SL, FHF and JZ, but there was a significant difference between CK. The soil organic matter content in treatment ZL was the highest, which was not significantly different from that in treatments DL, SL, FHF and JZ and significantly different from that in treatment CK. The soil pH value of treatment DL was the highest, which was not significantly different from that of treatments ZL and CK but significantly different from that of the other treatments. There was no significant difference in soil moisture content between treatments ZL, SL, FHF and CK, while there was a significant difference between treatments DL and JZ.

## Alpha diversity of the soil bacterial community

In this study, Illumina Miseq high-throughput sequencing was carried out in 6 different fertilization treatments of pakchoi. The results in Table 4 show that the OTU bacterial communities in each sample were treatment CK > FHF > DL > JZ > ZL > SL. There was no significant difference in the Chao1 index between treatment CK and treatments DL, ZL, FHF, JZ, but there was a significant difference between treatment CK and treatment SL. There was no significant difference in the Shannon indexes of the bacterial communities among treatments. The sequence depth index of the *nif*H gene in each sample ranged from 92.22% to 94.17%, which

**Table 3. Soil basic properties under different treatments.**

| Treatments | AN (mg/kg) | AP (mg/kg) | AK (mg/kg) | OM (g/kg) | pH | WC (%) |
|---|---|---|---|---|---|---|
| CK | 48.6±5.3[b] | 75.8±8.6[c] | 137.1±10.0[b] | 40.1±5.3[b] | 5.17±0.05[ab] | 27.50±0.66[a] |
| DL | 56.4±5.9[ab] | 93.2±9.2[bc] | 168.5±11.7[ab] | 50.1±4.8[a] | 5.36±0.08[a] | 25.23±0.73[b] |
| ZL | 65.1±6.5[a] | 112.8±7.3[a] | 176.3±13.5[a] | 51.6±5.8[a] | 5.31±0.03[ab] | 27.25±0.22[a] |
| SL | 58.6±3.4[ab] | 102.6±8.2[ab] | 160.1±15.1[ab] | 48.6±6.7[ab] | 4.78±0.07[c] | 27.63±0.11[a] |
| FHF | 52.4±5.2[ab] | 82.6±5.6[bc] | 145.6±11.3[ab] | 43.1±6.4[ab] | 5.13±0.12[b] | 27.44±0.73[a] |
| JZ | 54.6±5.3[ab] | 87.9±5.9[bc] | 148.9±7.6[ab] | 44.2±5.0[ab] | 4.82±0.04[c] | 25.72±0.25[b] |

OM, soil organic matter. pH, soil pH. AN, soil Alkali-hydrolyzed nitrogen. AP, soil available phosphorus. AK, soil available potassium. WC, soil water content. Note. The data are presented as the mean ± standard deviation (SD). Different letters in the same column mean significant difference at 0.05 level.

**Table 4. Analysis of soil bacterial community diversity under different fertilization treatments.**

| Treatment | Chao1 index | Shannon index | Good coverage (%) |
|---|---|---|---|
| DL | 3189.81±210.26[ab] | 9.38±0.02[a] | 93.77±0.36[ab] |
| ZL | 3252.57±443.93[ab] | 9.10±0.48[a] | 93.37±0.88[ab] |
| SL | 2864.32±211.32[b] | 9.03±0.36[a] | 94.17±0.46[a] |
| FHF | 3276.36±393.68[ab] | 9.13±0.63[a] | 93.15±0.80[ab] |
| JZ | 3225.52±268.01[ab] | 8.99±0.22[a] | 93.34±0.55[ab] |
| CK | 3752.93±245.23[a] | 9.74±0.26[a] | 92.22±0.46[b] |

The data are presented as the mean ± standard deviation (SD). Different letters in the same column mean significant difference at 0.05 level.

indicated that the coverage rate of each sample library was very high. The high probability of sequencing in each sample indicated that the sequencing result can reflect the actual situation of microorganisms in each sample. According to the Shannon-Wiener curve (Fig 2), the curves of all the samples tended to be flat, indicating that the amount of sequencing data is large enough to reflect the vast majority of microbial information in the samples.

PCA analysis of community composition structure at the generic level was carried out by R software. The results of the PCA community diversity analysis (Fig 3) showed that PC1 and PC2 explained 20.32% and 15.20% of the variation in the bacterial communities in pakchoi soil, respectively, and the cumulative total explained 35.52% of the total variables. The close distance between points DL and SL and ZL and FHF indicates the high similarity of the bacterial flora between the treatment points of DL and SL, ZL and FHF. The distance between all fertilization treatments and the control points was great, indicating that different fertilization treatments significantly changed the soil bacterial community structure.

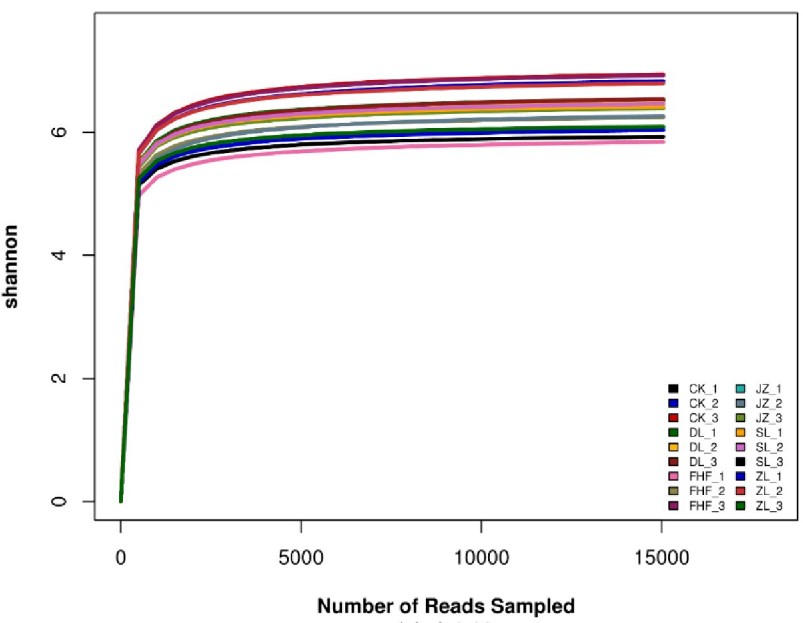

**Fig 2. Shannon-Wiener curves of soil bacterial communities treated with different fertilizers.**

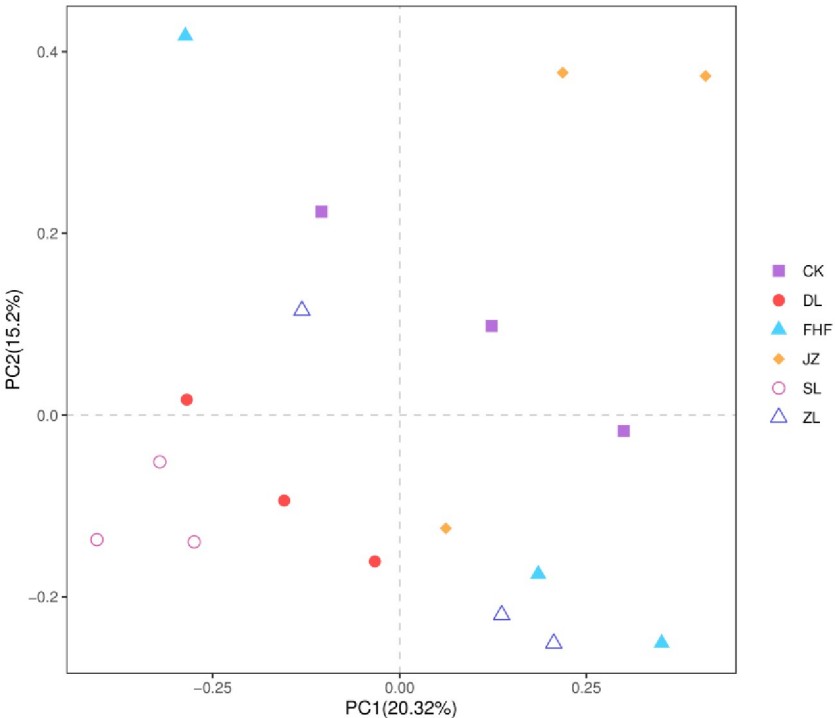

**Fig 3. PCA analysis of soil bacterial communities under different fertilization treatments.**

## Analysis of bacterial community composition at the phylum and genus levels

From the level of the middle phylum (Fig 4), Proteobacteria had the highest relative abundance of bacteria in the different fertilization treatments, with a relative abundance ranging from

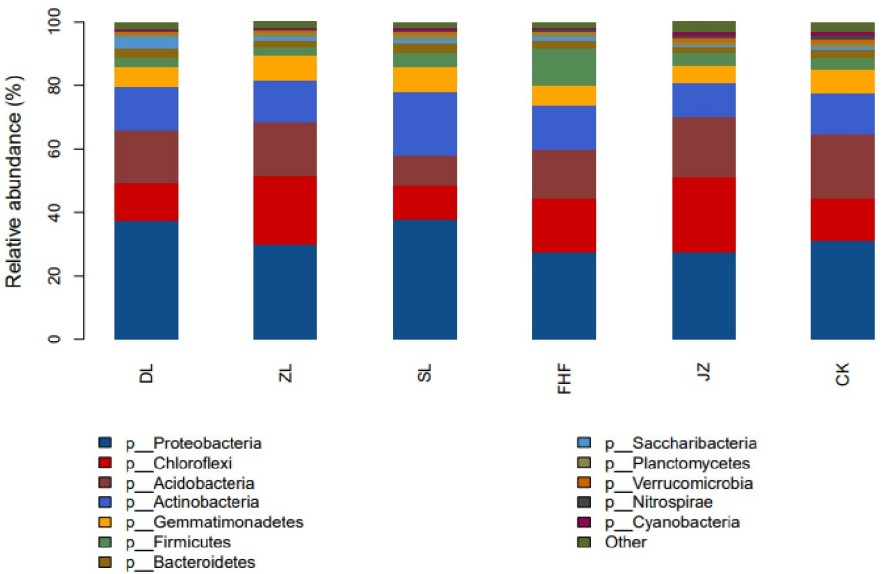

**Fig 4. Community composition of soil bacteria at the phylum level.**

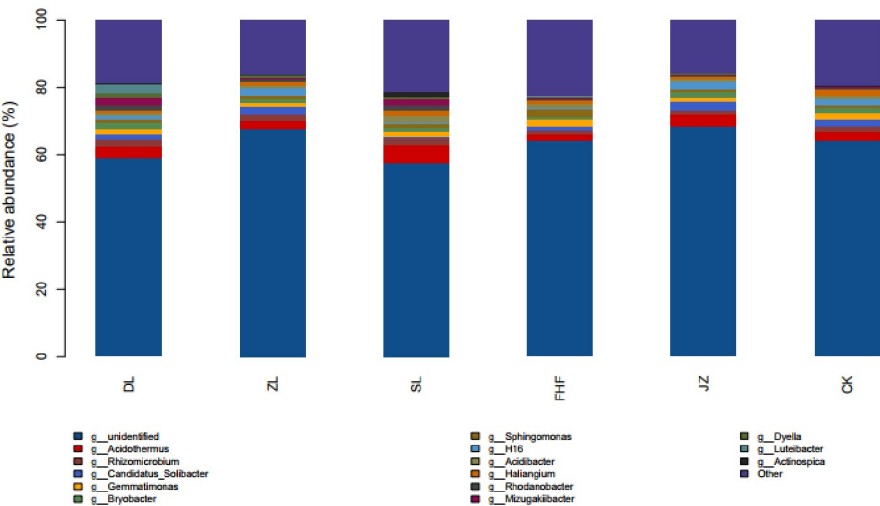

**Fig 5. Community composition of soil bacteria at the genus level.**

27.42% to 37.70%, followed by that of Chloroflexi (10.63%-24.07%), Acidobacteria (9.67%-20.27%), Actinobacteria (10.81%-20.09), Gemmatimonadetes (5.27%-7.99%), Firmicutes (2.84%-11.29%), and Bacteroidetes.1.83%-3.4%). The relative abundance of Acidobacteria in treatment CK was significantly higher than that in the other five treatments. The amount of Proteobacteria was higher in treatment DL than in the other five treatments. The relative abundance of Gemmatimonadetes in treatment ZL was obviously higher than that in the other five treatments. The relative abundance of Actinomycetes and Bacteroidetes in treatment SL was obviously higher than that in the other five treatments. The relative abundance of Firmicutes in treatment FHF was obviously higher than that in the other five treatments. The relative abundance of Chlorophora in treatment JZ was higher than that in the other 5 treatments.

At the genus level (Fig 5), the dominant bacteria genera under different treatments were *Acidothermus* (1.74%~4.78%), *Rhizomicrobium* (1.03%~2.39%), *Gemmatimonas* (1.21%~1.92%), *Candidatus_Solibacter* (1.09%~2.49%), *Bryobacter* (1.01%~1.43%), *Sphingomonas* (1.04%~2.11%), H16 (1.07%~2.16%), *Rhodanobacter* (1.42%~1.68%), *Mizugakiibacter* (1.76%~2.61%), *Dyella* (1.23%), *Luteibacter* (2.71%), *Haliangium* (1%~1.84%), *Acidibacter* (2.15%), *Actinospica* (1.27%), *Bryzomicrobium* (1.39%). The results showed that different fertilization treatments had a great influence on the species and relative abundance of soil bacteria.

## Relationship between soil bacterial communities and soil physical and chemical factors

Soil physical and chemical factors and bacterial communities were analyzed using RDA correlation analysis. RDA1 and RDA2 explained 56.46% and 26.07% (Fig 6), respectively, of the changes in bacterial communities in each sample. That is, all soil physical and chemical factors explained 82.53% of the changes in the bacterial communities. The correlation between soil moisture content and bacterial community was the highest, followed by that between soil pH, organic matter, available potassium and bacterial community, and the correlation between available phosphorus, alkali-hydrolyzed nitrogen and bacterial community was the smallest. The angle between the vector arrows of soil pH, organic matter, available potassium, available phosphorus and alkali-hydrolyzed nitrogen environmental factors was small and may have a synergistic effect.

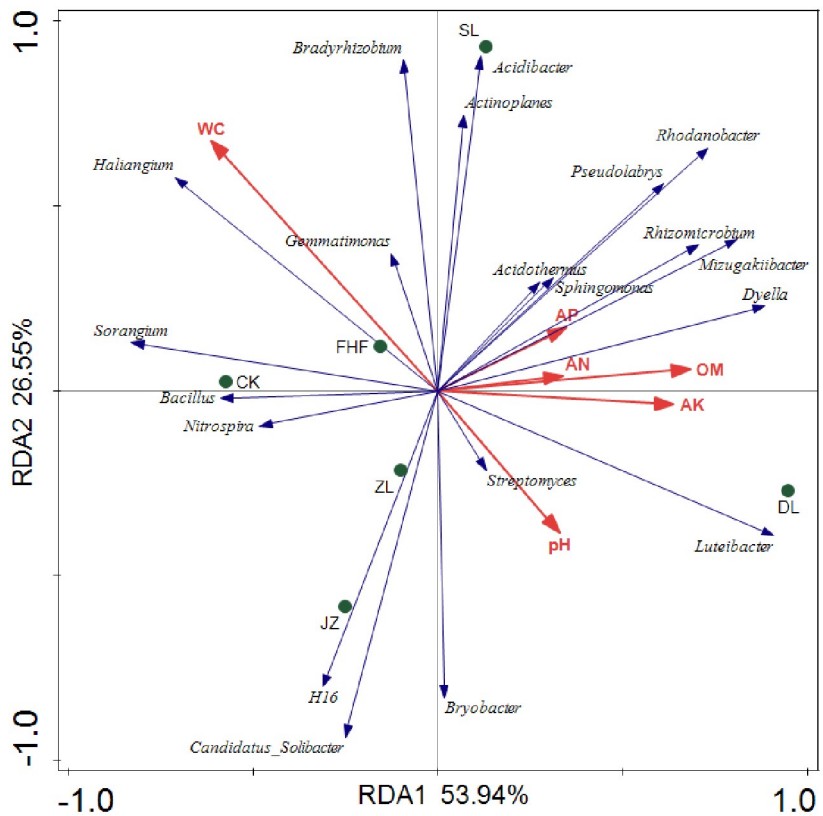

**Fig 6. RDA analysis of the soil bacterial community and soil properties.**

## Discussion

We studied the effects of inorganic fertilizer reduction combined with *Pennisetum giganteum* z.x.lin nitrogen-fixing biofertilizer on *Brassica chinensis* L. The results showed that the treatments had different effects on plant height, fresh weight per plant, chlorophyll content, soluble protein, soluble sugar, vitamin C and nitrate content. Among them, the 25% inorganic fertilizer reduction combined with *Pennisetum* giganteum z.x.lin mixed nitrogen-fixing biofertilizer had the best effect on plant height, fresh weight per plant, chlorophyll content, soluble protein, soluble sugar and vitamin C content, and these factors increased by 11.81%, 8.54%, 7.37%, 16.88%, 17.05%, 23.70%, respectively, and the nitrate content decreased by 53.85% in this treatment compared with that in the conventional fertilization treatment. Studies have shown that biofertilizer has a great effect on maize yield [36]. Biofertilizer treatment can improve the chlorophyll content of oat leaves and promote the accumulation of total nitrogen in stems, leaves and ears of grain [37].

Soil microorganisms are an important factor affecting soil ecological processes. These microorganisms play an important role in soil formation, the biogeochemical cycle of ecosystems, the degradation of pollutants and the maintenance of groundwater quality [38–39]. Soil microbial diversity is considered to be an important factor in maintaining soil health [40]. The results of this study showed that the dominant phyla of rhizosphere soil bacteria were Proteobacteria, Chloroflexi, Acidobacteria, Actinobacteria, Firmicutes, Gemmatimonadetes and Bacteroidetes. The results are similar to those obtained in previous studies on different types of farmland soil. The difference between those studies and this study is that the relative

abundance of each dominant group varies greatly because the species and abundance of the dominant groups are influenced by soil type, texture and crop varieties [41–42]. Compared with no fertilization and only the application of inorganic fertilizer, the application of *Pennisetum* giganteum z.x.lin mixed nitrogen-fixing biofertilizer combined with inorganic fertilizer significantly improved the abundance of Proteobacteria and Actinomycetes in the soil. Zhang et al. [43] found that microbial organic fertilizer increased the abundance of soil Proteobacteria. Wang et al. [44] found that fertilization and straw mulching significantly increased the relative abundance of Actinomyces. Our results are consistent with those of these authors. Previous studies have shown that Proteobacteria are gram-negative bacteria and are eutrophic bacteria. These bacteria have a positive correlation with nutrient content [45–46], which is critical to the global carbon, nitrogen and sulfur cycles [47], and they play a very important role in biological nitrogen fixation, biological control and plant growth promotion [48]. Actinomycetes can produce a variety of secondary metabolites (antibiotics) and extracellular enzymes, which play an important role in the defense of plant diseases [49]. The increase in the abundance of these two types of bacteria plays an important role in improving soil quality, increasing soil nutrient content, enhancing stress resistance and promoting plant growth and is conducive to the sustainable development of soil microecology. The increase in nutrient content in soil was assumed to be due to stimulation of the growth of Proteobacteria and Actinomycetes by applying the *Pennisetum giganteum* z.x.lin mixed nitrogen-fixing bacterial fertilizer combined with inorganic fertilizer, which led to their increase in abundance in the soil.

With the development of modern agriculture in China, the inappropriate application of inorganic fertilizers has caused increasingly serious harm to the agricultural ecological environment. In recent years, with the increasing attention of society on the protection of agricultural ecological environments, research on replacing inorganic fertilizers with biofertilizer has attracted much attention [50]. Han et al. [51] showed that the content of soil organic matter, total nitrogen, total phosphorus, total potassium, available phosphorus and available potassium increased by 42.2%, 58.8%, 8%, 12.6%, 37.2% and 40.2%, respectively, when treated with rhizobia and PGPR bacterial fertilizer. Pang et al. [52] showed that soil organic matter and quick-effect N, P and K contents were significantly higher than those in a control treatment without adding microbial agents. The results of this study showed that compared with no fertilization and only the application of inorganic fertilizer, the combined application of inorganic fertilizer and *Pennisetum giganteum* z.x.lin mixed nitrogen-fixing biofertilizer could improve the physical and chemical properties of the soil to a certain extent. When a 25% reduction in chemical fertilizer was combined with *Pennisetum giganteum* z.x.lin mixed nitrogen-fixing biofertilizer, the content of alkali-hydrolyzed nitrogen, available phosphorus, available potassium and organic matter in the soil increased most obviously, and the soil pH also improved to a certain extent. Changes in soil nutrients change the structure and functional diversity of microbial communities [53]. The growth, activity and functional diversity of the soil microbial community were affected by various soil physical and chemical properties, including soil pH, total nitrogen (TN), soil organic carbon (SOC) and soil enzyme activity. Under fertilization, pH is the main driving factor changing microbial communities such as bacteria, fungi, archaea and protozoa. TN, SOC, enzyme activity and other factors are important contributors to the composition of different microbial communities [54–57]. Soil bacteria are essential for maintaining soil fertility and ecosystem functions and are often sensitive to fertilizer inputs [58]. Many studies have shown that fertilization changes the soil fertility and nutrient content, such as organic carbon [59] and total nitrogen [60–61], directly driving the transformation of soil microbial communities and increasing or decreasing the diversity of the microbial community. Fertilization also indirectly affects soil microorganisms by changing soil properties [62]. Soil microorganisms can secrete active growth substances such as auxin, cytokinin, and zeatin

to promote plant growth [63]. Therefore, under fertilization, soil physical and chemical properties and microorganisms interact to jointly promote the growth of plants, improving the yield of plants. The results of this study showed that soil organic matter, alkali-hydrolyzed nitrogen, available phosphorus, available potassium, pH and water content all had certain effects on the changes in the soil bacterial community. Among the factors, soil water content was the main factor affecting the change of bacterial community. Soil pH is also an important index affecting bacterial community structure [64–65]. However, pH was not the main factor affecting the structure of the bacterial community in this study. This result may have been due to the differences in the soil environments between different research areas or the relatively small differences in the pH values of the soil samples among the different fertilization treatments, which did not meet the threshold of changing bacterial communities and thus did not have enough impact on the soil bacterial community. The effects of fertilizer application on soil bacterial diversity and community composition were different in the ecosystems [66–70]. At present, there is contradictory information on the impact of biofertilizer on soil microbial diversity, which may be related to the type of fertilizer, the amount and duration of application, soil type and utilization mode. Therefore, the influence of other factors on soil microbial diversity needs to be further studied.

## Conclusions

In conclusion, when the ratio of inorganic fertilizer and *Pennisetum giganteum* z.x.lin mixed nitrogen-fixing biofertilizer is appropriate, it can promote the growth of pakchoi, improve the nutritional quality, soil fertility and bacterial community of pakchoi. When the reduction ratio of chemical fertilizer is large, it can also improve the quality of pakchoi. However, the reduced ratio hinders the growth of pakchoi due to the lack of nutrient supply. When the reduction ratio of chemical fertilizer is small, the nutrient supply is sufficient, but the change in the soil environment leads to a reduction in beneficial bacteria in the soil, thus hindering the growth of pakchoi and reducing its quality. Therefore, considering factors such as the growth and quality of pakchoi and its soil environment, the effect of a 25% reduction of chemical fertilizer and the application of *Pennisetum giganteum* mixed nitrogen-fixing bacterial manure is the best. The results of this study can provide an important theoretical basis for the rational reduction of chemical fertilizer and the scientific and reasonable application of *Pennisetum giganteum* mixed nitrogen-fixing bacterial manure.

## Supporting information

**S1 File.**
(DOCX)

## Author Contributions

**Data curation:** Yulei Jia, Zhen Liao.

**Formal analysis:** Yulei Jia.

**Investigation:** Yulei Jia, Zhen Liao, Huifang Chew, Zhanxi Lin.

**Methodology:** Yulei Jia, Zhen Liao, Huifang Chew, Lifang Wang, Biaosheng Lin, Chaoqian Chen, Guodong Lu, Zhanxi Lin.

**Supervision:** Yulei Jia, Zhanxi Lin.

**Writing – original draft:** Zhen Liao.

**Writing – review & editing:** Yulei Jia, Huifang Chew, Zhanxi Lin.

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
