## [Decision Letter · Decision Letter 0]

22 Oct 2019

PONE-D-19-26117

Effect of chemical fertilizer reduction and co-application with Pennisetum sp. mixed nitrogen-fixing bacterial fertilizer on growth, quality, soil fertility and bacterial community of pakchoi (Brassica chinensis L.)

PLOS ONE

Dear Mr Jia,

Thank you for submitting your manuscript to PLOS ONE. After careful consideration, we feel that it has merit but does not fully meet PLOS ONE’s publication criteria as it currently stands. Therefore, we invite you to submit a revised version of the manuscript that addresses the points raised during the review process.

We would appreciate receiving your revised manuscript by Dec 06 2019 11:59PM. To enhance the reproducibility of your results, we recommend that if applicable you deposit your laboratory protocols in protocols.io, where a protocol can be assigned its own identifier (DOI) such that it can be cited independently in the future. For instructions see: http://journals.plos.org/plosone/s/submission-guidelines#loc-laboratory-protocols

We look forward to receiving your revised manuscript.

Kind regards,

Balasubramani Ravindran, Ph.D

Academic Editor

PLOS ONE

Journal Requirements:

'This work was supported by the Central Leading Local Science and Technology Development Special Fund Project of China under grant number 2018L3003; the Ministry of Agriculture, Grass and Ecology Industry and the Ministry of Construction and Collaborative Innovation Center, China under grant number [2018]126'

'the Central Leading Local Science and Technology Development Special Fund Project of China under grant number 2018L3003'

Additional Editor Comments (if provided):

Reviewers' comments:

Reviewer's Responses to Questions

**Comments to the Author**

1. Is the manuscript technically sound, and do the data support the conclusions?

Reviewer #1: Yes

Reviewer #2: Partly

2. Has the statistical analysis been performed appropriately and rigorously? 

Reviewer #1: Yes

Reviewer #2: No

3. Have the authors made all data underlying the findings in their manuscript fully available?

Reviewer #1: Yes

Reviewer #2: Yes

4. Is the manuscript presented in an intelligible fashion and written in standard English?

Reviewer #1: Yes

Reviewer #2: No

5. Review Comments to the Author

Reviewer #1: The authors investigated the effects of chemical fertilizer reduction and co-application with Pennisetum sp. Mixed nitrogen-fixing bacterial fertilizer on growth, quality, soil fertility and bacterial community of pakchoi. They found that only 25% chemical fertilizer reduction combined with Pennisetum sp. mixed nitrogen-fixing bacterial manure have the best effect. However, in its current state the authors spend too much time describing the results and too little time discussing their implications. Another issue is the language. Given the large number of writing issues, the author are strongly encouraged to seek further help from a native English speaker to improve the writing.

Below are minor, but more specific comments.

You need to simplify and modify your title to make it concise and clear.

Abstract:

L2-5, is it a sentence?

L21 most obviously ?

Introduction:

L23: improve soil ? please modify it.

Discussion

L19-20: presumably due to the application of chemical fertilizer will cause soil

acidification??? There are many mistakes like this. Please check you sentences carefully.

What is the relationship between soil physical and chemical properties and yield of pakchoi (Brassica chinensis L.)? Please discuss it.

How did you draw the conclusion that 25% chemical fertilizer reduction combined with Pennisetum sp. mixed nitrogen-fixing bacterial manure was the best practice to improve the growth and quality of pakchoi (Brassica chinensis L.)? To the extent, the conclusions section is a complete repetition of the main results. Please compare the difference of treatments and discuss it.

Please discuss how the soil properties affect the bacterial community structure and the growth of pakchoi (Brassica chinensis L.). Add latest references to enrich your paper.

Reviewer #2: Comments to authors

I have reviewed the manuscript titled “Effect of chemical fertilizer reduction and co-application with Pennisetum sp. Mixed nitrogen-fixing bacterial fertilizer on growth, quality, soil fertility and bacterial community of pakchoi (Brassica chinensis L.).”

From the beginning, I failed to grasp what the title was looking into, Pennisetum sp mixed nitrogen fixing bactierial fertilizer to me is meaningless. Pennisetum is a type of millet and how its used to prepare a fertilizer was not at all clear from the title to the manuscript contents itself. Overally, the experimental design does not allow for the repetition of this study at all, as there are a lot of variables involved and a lot of unclear treatment compositions. I do not feel this manuscript can be accepted in its form, but maybe with careful major revision it can be considered. The English is very bad and this needs to be edited by a profession English editor. Lack of continuous line numbers and page numbers also make it a challenge to make specific comments to the manuscript.

I have made specific comments below.

Title

I cannot get the meaning of what the title is looking into, what is Pennisetum sp. Mixed what what. How do you mix a millet with N fixing bacteria to create a fertilizer?

Abstract

What is hm2, what unit is this?

Line 5, including CK, what is this CK?

The 1st sentence reading “to explore the effects… is not complete, what was done to explore these. At least give one background statement.

Abstract is just too long, and the last sentence “the results of this study provide theoretical and scientific….what does this mean.

Introduction

The section fails to justify why this study was undertaken, what other researchers have done and what this study will contribute. Its common knowledge that inorganic fertilizers are detrimental to the soil, so, what does this study seek to address, apart from presenting an alternative organic fertilizer.

Avoid using plurals of words, such as, Indicators (line 5); promotes (line 8); researches (line 18) among others.

Line 7, how does genetic diversity promote decomposition, this is not clear.

Line 11, chemical fertilizer, do you mean inorganic, rather use this term.

Line 19, what is a microbial fertilizer, this is a new term.

Line 29, small and large, meaning?

Line 35, Pennisetum sp. Mixed… I can’t understand this part.

Line 42, you need a clear Objective, it has to be SMART.

Materials and methods

This section is poorly written and does not allow the repetition of this study. Some of the sticking points are: which Pennisetum species, nutrient solution was brown sugar (meaning); how do u isolate bacteria from Pennisetum?; fermentation was end if the material became soft and dark (this is not a quantifiable character, how do you repeat this?).

250-350 g of mushroom waste, why mushroom, and which species.

Line 16, Material, use a better term.

Line 17, 2 tested N fixing strains, give their full names, and how do you isolate these from a grass, Pennisetum.

Line 18, preserved in the laboratory, not authors lab.

Which buy from local market, correct gramma

Line 19-25; Chemical fertilizer….its not clear, I cant follow what was done here.

Line 31, how was the isolation done.

What is hm2

Table 1m give the actual names of the species used.

On soil chemical analysis, give the actual methods used to extract. Eg, available P was measured by calorimetry, how was the P extracted in the 1st place and provide references.

Data analysis, Factors with P < 0.05; what do you mean, which factors. Write this section clearly.

Results

Morphology, what is this, I see nothing relating to plant morphology. A parameter like number of leaves is very useless to measure as this is a GxE interaction.

Avoid referring to your treatments as A, BC.

Table 2, +/- what value.

Table 3, that footnote is a repletion of stuff presented before.

Discussion fails to link the results and explain what could have caused the responses. Avoid repeating results.

What is Naked oat leaves.

Conclusion

Best effect, what effect are you referring to.

Enhancing its quality, what is this, be specific to the parameter used to measure this.

…………………………………..END……………………….

6. PLOS authors have the option to publish the peer review history of their article (what does this mean?). If published, this will include your full peer review and any attached files.

Reviewer #1: No

Reviewer #2: No

---

## [Author Response · Author response to Decision Letter 0]

11 Dec 2019

Reviewer #1:

Abstract

L2-5, is it a sentence? 

Yes, editor. It is a sentence. I use the Pennisetum giganteum z.x.lin mixed nitrogen-fixing biofertilizer. I test the growth, quality, physical and chemical properties of the soil and the bacterial community of the soil. I have revised this sentence in my paper. Please check it.

L21 most obviously ?

Dear editor, my most obviously is mean the

---

## [Decision Letter · Decision Letter 1]

23 Jan 2020

Effect of Pennisetumgiganteum z.x.lin mixed nitrogen-fixing bacterial fertilizer on the growth, quality, soil fertility and bacterial community of pakchoi (Brassicachinensis L.)

PONE-D-19-26117R1

Dear Dr. Yulei Jia,

We are pleased to inform you that your manuscript has been judged scientifically suitable for publication and will be formally accepted for publication once it complies with all outstanding technical requirements.

With kind regards,

Balasubramani Ravindran, Ph.D

Academic Editor

PLOS ONE

Reviewers' comments:

Reviewer's Responses to Questions

**Comments to the Author**

1. If the authors have adequately addressed your comments raised in a previous round of review and you feel that this manuscript is now acceptable for publication, you may indicate that here to bypass the “Comments to the Author” section, enter your conflict of interest statement in the “Confidential to Editor” section, and submit your "Accept" recommendation.

Reviewer #1: All comments have been addressed

Reviewer #2: All comments have been addressed

2. Is the manuscript technically sound, and do the data support the conclusions?

Reviewer #1: Yes

Reviewer #2: Yes

3. Has the statistical analysis been performed appropriately and rigorously? 

Reviewer #1: Yes

Reviewer #2: Yes

4. Have the authors made all data underlying the findings in their manuscript fully available?

Reviewer #1: Yes

Reviewer #2: No

5. Is the manuscript presented in an intelligible fashion and written in standard English?

Reviewer #1: Yes

Reviewer #2: Yes

6. Review Comments to the Author

Reviewer #1: The authors generally did a good job in addressing the comments, concerns and suggestions of the editor and both reviewers. Specifically, the English has been corrected. I think the manuscript will be acceptable after minor revision.

Given the large number of writing issues identified, the authors are strongly encouraged to seek further help from a native English speaker to improve the writing and make it more concise. Why not change you FIg.1 to color?

Reviewer #2: Abstract

The experiment was divided into 6 treatments, rather, the experiment composed of 6 treatments.

I do not understand the reason for using terms like CK, DL, ZL to define treatments, what is the link, I understand the treatments are described in brackets but what is the use of using these letters.

Introduction

Line 42: participating in material cycling processes, Rather, participating in Nutrient cycling.

Pennisetum giganteum z.x.lin belongs to Angiospermae, Monocotyledon, Gramineae and Pennisetum. This statement is not complete, if it belongs to this, and so what?

Materials and methods

Experimental design: what instructed the application rate of the organic fertilizer in Table 1.

Sampling and analysis, this section is too crowded, rather use sub sections and indicate references for each method used. Plant height of each plant…this is grammatically not correct, just rather say plant height was measured.

Indicate the package used for data analysis.

Results

Table 3, indicate with a footprint what the small letters indicate, what does AN, AP stand for, these need to be described also in the Table. What does the +/- indicate also. Indicate in all tables.

Bacterial community analysis: Line 26 onwards, can this information not be presented in anoher way, its too crowded to follow.

Conclusion

This statement is not making sense to me.

In conclusion, because the ratio of reduced fertilizer and combined Pennisetum

32 giganteum z.x.lin mixed nitrogen-fixing biofertilizer is suitable, it can promote the

33 growth and quality of pakchoi, soil quality and soil bacterial community structure.

7. PLOS authors have the option to publish the peer review history of their article (what does this mean?). If published, this will include your full peer review and any attached files.

Reviewer #1: No

Reviewer #2: Yes: Hupenyu Allan Mupambwa

---

## [Editor Report · Acceptance letter]

27 Jan 2020

PONE-D-19-26117R1 

Effect of *Pennisetum giganteum* z.x.lin mixed nitrogen-fixing bacterial fertilizer on the growth, quality, soil fertility and bacterial community of pakchoi (*Brassica chinensis* L.) 

Dear Dr. Jia:

I am pleased to inform you that your manuscript has been deemed suitable for publication in PLOS ONE. Congratulations! Your manuscript is now with our production department. 

With kind regards,

on behalf of

Dr. Balasubramani Ravindran 

Academic Editor

PLOS ONE